# Silencing of the 20S proteasomal subunit-α6 triggers full oogenesis arrest and increased mRNA levels of the selective autophagy adaptor protein p62/SQSTM1 in the ovary of the vector *Rhodnius prolixus*

**Allana Faria-Reis**[1], **Samara Santos-Araújo**[1], **Jéssica Pereira**[1], **Thamara Rios**[1], **David Majerowicz**[2,3,4], **Katia C. Gondim**[1,4], **Isabela Ramos**[1,4]*

**1** Instituto de Bioquímica Médica Leopoldo de Meis, Universidade Federal do Rio de Janeiro, Brazil, **2** Departamento de Biotecnologia Farmacêutica, Faculdade de Farmácia, **3** Programa de Pós-Graduação em Biociências, Universidade do Estado do Rio de Janeiro, **4** Instituto Nacional de Ciência e Tecnologia em Entomologia Molecular–INCT-EM/CNPq

* isabela@bioqmed.ufrj.br

## Abstract

The high reproductive rates of insects contribute significantly to their ability to act as vectors of a variety of vector-borne diseases. Therefore, it is strategically critical to find molecular targets with biotechnological potential through the functional study of genes essential for insect reproduction. The ubiquitin-proteasome system is a vital degradative pathway that contributes to the maintenance of regular eukaryotic cell proteostasis. This mechanism involves the action of enzymes to covalently link ubiquitin to proteins that are meant to be delivered to the 26S proteasome and broken down. The 26S proteasome is a large protease complex (including the 20S and 19S subcomplexes) that binds, deubiquitylates, unfolds, and degrades its substrates. Here, we used bioinformatics to identify the genes that encode the seven α and β subunits of the 20S proteasome in the genome of *R. prolixus* and learned that those transcripts are accumulated into mature oocytes. To access proteasome function during oogenesis, we conducted RNAi functional tests employing one of the 20S proteasome subunits (Prosα6) as a tool to suppress 20S proteasomal activity. We found that Prosα6 silencing resulted in no changes in TAG buildup in the fat body and unaffected availability of yolk proteins in the hemolymph of vitellogenic females. Despite this, the silencing of Prosα6 culminated in the impairment of oocyte maturation at the early stages of oogenesis. Overall, we discovered that proteasome activity is especially important for the signals that initiate oogenesis in *R. prolixus* and discuss in what manner further investigations on the regulation of proteasome assembly and activity might contribute to the unraveling of oogenesis molecular mechanisms and oocyte maturation in this vector.

**Data Availability Statement:** All relevant data are within the manuscript and its Supporting Information files.

**Funding:** This research was funded by the following grants: Jovem Cientista do Nosso Estado (JCNE)-Fundação de Amparo à Pesquisa do Estado do Rio de Janeiro (FAPERJ) (www.faperj.br/), Instituto Nacional de Ciência e Tecnologia em Entomologia Molecular (INCT-EM)-Conselho Nacional de Desenvolvimento Científico e Tecnológico (CNPq) (http://cnpq.br/) and Coordenação de Aperfeiçoamento de Pessoal de Nível Superior (CAPES) (www.capes.gov.br/) to I. R. The funders had no role in study design, data collection and analysis, decision to publish, or preparation of the manuscript.

**Competing interests:** The authors declare no conflict of interest.

## Author summary

In underdeveloped nations, the high frequency of neglected vector-borne illnesses like Chagas disease imposes significant health and financial burdens. Throughout history, the major efficient management strategies for such illnesses have relied on vector population control and continue to do so now. Although early efforts to understand the biology of specific vectors led to significant advances in the creation of management strategies for vector-borne diseases, studies regarding the intricate physiology of local vector species were hindered by the growing use of insecticide-based tools. Currently, the necessity to rely on in-depth species-specific vector biology has been highlighted once again by the growing threat of pesticide resistance and climate change (which can expand endemic areas). Disrupting molecular processes or attacking the metabolic targets required to generate viable eggs is one method of managing vector populations. Here, we present discoveries regarding the molecular basis of oogenesis of the insect vector of Chagas Disease *Rhodnius prolixus*. We found that proteasome activity is important for the unfolding of the oogenesis program, being essential for the reproductive success of this vector.

## Introduction

Early in the 20th century, the strictly hematophagous hemipteran *Rhodnius prolixus* was used as a model for the discovery of several fundamental aspects of insect physiology [1–6]. This insect is also significant for human health as the vector of the neglected tropical disease named Chagas Disease [7], which is endemic in Central and South America [8]. The Chagas disease already affects more than 8 million individuals, and predictions show that climate change and globalization will cause its endemic zone to expand significantly more [9].

The reproductive systems of female insects have been extensively studied due to their potential as targets for insect control. Basically, the maturing oocytes internalize yolk proteins via receptor-mediated endocytosis after the fat body produces and secretes vitellogenin (Vg) and other precursors of yolk proteins, and this is a crucial phase in the female reproduction process in most insects [10,11]. The adult female reproductive system of *R. prolixus* is composed of two ovaries, each containing 7 telotrophic ovarioles. Oogenesis has three phases known as pre-vitellogenesis, vitellogenesis and choriogenesis. In the vitellarium, pre-vitellogenic and vitellogenic oocytes are connected to the nurse cells in the tropharium by a nutritive cord, which allows the transport of macromolecules, mRNA, and protein to the developing oocytes [12]. Vitellogenin (Vg), the main yolk protein precursor, is mostly provided by the fat body, but it is also synthesized within the follicle cells and delivered to the oocyte in the late phase of growth [13,14]. Once vitellogenesis is complete, the trophic cord is severed, the chorion is synthesized and secreted, and the mature egg becomes ready to be fertilized and laid in the environment.

A significant degradative mechanism that contributes to the precise controls that keep the steady-state levels of proteins in the cell is the ubiquitin-proteasome system (UPS) [15,16]. In this system, E1-activating, E2-conjugating, and E3-ligase enzymes work in an organized manner to covalently link ubiquitin to proteins that are intended for breakdown and deliver them to the proteolytic complex 26S proteasome [17–20]. The 26S proteasome is a large multimeric protease complex that binds, deubiquitylates, and unfolds its substrates before completing their degradation. The 26S complex is formed by two sub-complexes, the 19S regulatory particle, and the 20S core particle [21,22]. The 20S core particle is made up of two varieties of heptameric rings organized in a cylindrical α-β-β-α four-layered barrel structure [23–25]. Both the α- and β-rings in eukaryotes are heteroheptamers made up of seven unique but related

subunits. These subunits assembly is not a self-sufficient process, rather, it is aided by many chaperones that act as molecular intermediaries to avoid the biogenesis of defective assembly products [25–30]. The 20S particle proteolytic active sites are located in the β1, β2, and β5 subunits, and each site preferentially cleaves after specific amino acid residues [31].

The UPS controls numerous pathways and is an essential element of the cellular proteostasis network in eukaryotic cells [21]. Thus, it is not unexpected that numerous human diseases are linked to the dysfunction of the ubiquitination system or the proteasome's proteolytic activity, such as Alzheimer's and Parkinson's disease [32,33] as well as cardiovascular disorders [34]. Although the essential participation of proteasome activity is widespread, functional studies have demonstrated the particular role of proteasomes in the reproduction biology of several models. In the nematode *Caenorhabditis elegans*, subunits of the 19S regulatory complex participate in sex determination and oocyte maturation [35]. In the goldfish (*Carassius auratus*) it was demonstrated that modifications of proteasomal subunits occur during oogenesis and that these modifications may be involved in the regulation of oocyte maturation [36]. In the study models *Xenopus laevis* and mice, it has been extensively demonstrated that the UPS participates in oocyte meiotic maturation, activation, and fertilization [37–43], as well as degradation of maternal proteins during the maternal to zygotic transition [44,45]. In insects, functional studies were performed in the fruit fly *Drosophila melanogaster* and showed that physiological proteasome activity is required to ensure normal progression of oogenesis [46]. In the brown planthopper, *Nilaparvata lugens*, the knockdown of different 26S proteasome subunits resulted in impaired oocyte maturation [47,48]. In the kissing bug *R. prolixus*, RNAi knockdown of the ubiquitin enzymes E1 and E2 resulted in impaired oogenesis and embryo lethality [49].

In this work, using bioinformatics, we identified the genes encoding the seven α and β subunits of the 20S proteasome in the *R. prolixus* genome and noticed that their transcripts are maternally accumulated in the mature oocytes transcriptome [50]. To test the role of the proteasome activity during oogenesis, we performed RNAi functional studies using one of the 20S proteasome α subunits (Prosα6) as a tool to inhibit general proteasome activity. Although the silencing of Prosα6 resulted in decreased proteasomal activity, no major changes were observed in the amounts of triacylglycerol (TAG) in the fat body and yolk protein availability in the hemolymph. However, a full impairment of oocyte maturation was observed at the early stages of oogenesis, directly impacting the reproduction capacity of this model. Overall, we discovered that *R. prolixus* oogenesis is highly dependent on proteasome activity, and we highlight how further study into the regulation of proteasome assembly and activity may shed light on the molecular processes underlying oogenesis and, consequently, reproduction in this vector.

## Material and methods

### Ethics statement

All animal care and experimental protocols were approved by guidelines of the institutional care and use committee (Committee for Evaluation of Animal Use for Research from the Federal University of Rio de Janeiro, CEUA-UFRJ #01200.001568/2013-87, order number 149–19), under the regulation of the national council of animal experimentation control (CONCEA). Technicians dedicated to the animal facility conducted all aspects related to animal care under strict guidelines to ensure careful and consistent animal handling.

### Gene Identification and phylogenetic analysis

The genomes of *R. prolixus* [51], *D. melanogaster* [52], the mosquito *Aedes aegypti* [53], the bee *Apis mellifera* [54], the postman butterfly *Heliconius melpomene* [55], the beetle *Tribolium*

*castaneum* [56], the aphid *Acyrthosiphon pisum* [57], the whitefly *Bemisia tabaci* [58], the bed bug *Cimex lectularius* [59], the termite *Zootermopsis nevadensis* [60], and the water flea *Daphnia magna* [61] were explored. All proteins containing the Pfam domain [62] PF00227 (proteasome subunit) were obtained from the EnsemblMetazoa database [63] using the BioMart tool [64]. The primary sequences were aligned with the MUSCLE tool [65] and the phylogenetic analysis was performed by the maximum likelihood method [66] with 1000 bootstrap repetitions in the MEGA X software [67]. The dendrograms were visualized in Figtree software. *D. magna* sequences were included as an external group.

## Insects

Insects were maintained at a 28 ± 2°C controlled temperature, relative humidity of 65–85%, and 12/12 h light and dark cycles. All females used in this work were obtained from our insectarium where mated females are fed for the first time (as adult insects) in live-rabbit blood 14 to 21 days after the 5th instar nymph to adult ecdysis. After the first blood feeding, all adult insects in our insectarium are fed every 21 days. For all experiments, mated (females that produced a full batch of viable eggs during the first cycle of feeding as adults) and fully gorged (allowed to feed at free demand, usually gaining 6–7 times the insect's initial body weight in 20–30 min) females of the second or third blood feeding were used, and dissections were carried out on different days after the blood meal depending on the experiment.

## Dissection of organs and ovarian follicles

All dissections were carefully performed in phosphate-buffered saline (PBS, 137 mM NaCl, 2.7 mM KCl, 10 mM $Na_2HPO_4$, and 1.8 mM $KH_2PO_4$, pH 7.4), using fine tweezers and dissecting scissors under the stereomicroscope at day 7 after the blood meal. For RNA extractions the fat body and midgut were dissected as previously reported and midguts were carefully rinsed in PBS to wash away any residue of blood meal contents [68–70]. Both ovaries of each female were dissected. Ovary parts were dissected according to Huebner and Injeyan, 1980 [71], and the structures (tropharium, previtellogenic and vitellogenic follicles, and chorionated oocytes) were classified by length and morphology according to [72,73].

## Extraction of total RNA and cDNA synthesis

All samples were homogenized in TRIzol reagent (Invitrogen) for total RNA extraction. Reverse transcription reaction was carried out using the High-Capacity cDNA Reverse Transcription Kit (Applied Biosystems) using 1 µg of total RNA after RNase-free DNase I (Invitrogen) treatment, Multiscribe Reverse Transcriptase enzyme (2.5 U/µL) and random primers for 10 min at 25°C followed by 2 hours of incubation at 37°C. As a control for the DNAse treatment efficiency, we performed control reactions without the enzyme followed by testing the capacity of amplification by PCR.

## PCR/RT-qPCR

Specific primers for the *R. prolixus* Prosα6 sequence were designed to amplify a fragment of 121 bp in a PCR using the following cycling parameters: 5 min at 95°C, followed by 35 cycles of 30 s at 95°C, 30 s at 50°C and 60 s at 72°C and a final extension of 15 min at 72°C. Amplifications were observed in 2% agarose gels. Reverse transcription quantitative real-time PCR (RT-qPCR) was performed in a StepOne Real-Time PCR System (Applied Biosystems) using SYBR Green PCR Master Mix (Biosystems) under the following conditions: 10 min at 95°C, followed by 40 cycles of 15 s at 95°C and 45 s at 60°C. qPCR amplification was performed

using the specific primers described in S1 Table. All primers were used at a final concentration of 0.2 μM. The cDNAs were diluted 10x and used in the reactions. To exclude nonspecific amplification, blank reactions replacing de template (cDNA) for water were performed in all experiments. The relative expressions were calculated using the delta $C_t$ (cycle threshold) obtained using the reference gene 18S ribosomal RNA (18S) (RPRC017412) and calculated $2^{-dCt}$ [74]. According to the minimum information for publication of quantitative RT-qPCR experiments (MIQE) Guidelines, normalization against a single reference gene is acceptable when the investigators present clear evidence that confirms its stable expression under the experimental conditions [75]. S1 Fig shows the invariant expression of 18S in our experimental conditions.

## RNAi silencing

dsRNA was synthesized by MEGAScript RNAi Kit (Ambion Inc) using primers for specific *R. prolixus* Prosα6 gene amplification with the T7 promoter sequence (S1 Table). Unfed adult females were injected between the second and third thoracic segments using a 10 μl Hamilton syringe with 1 μg dsRNA (diluted in 1μl of water) and fed 2 days later. Knockdown efficiency was confirmed by qPCR at 7 days after the blood meal in the ovary, fat body, and midgut. The bacterial *MalE* gene was used as a control dsRNA [76]. Adult females injected with dsRNA were fed and transferred to individual vials. The mortality rates and the number of eggs laid by each individual were recorded weekly and every 3 days, respectively.

## Production of anti-*R. prolixus* Prosα6 antibodies

Specific polyclonal antibodies for the single isoform of *R. prolixus* Prosα6 were raised commercially by GeneScript. Rabbits were immunized with a 14-amino acid peptide (NH2-RDSSPNDIELNNKN-COOH) derived from the predicted Prosα6 sequence. Pre-immune serum was tested to monitor the absence of cross-reactivity with the samples. Anti-Prosα6 antiserum was used for immunoblotting.

## Determination of protein content

The total amount of protein was measured by a microtechnique of the Lowry (Folin) method [77] adapted to a 96 well plate using as standard control ranging from 1–5 μg of BSA in a E-MAX PLUS microplate reader (Molecular Devices) using SoftMax Pro 7.0 as software.

## Immunoblotting

Ovaries were dissected and homogenized using a glass/Teflon potter Elvehjem homogenizer in PBS. The homogenates, containing 60 μg of total protein, were separated by a 12% SDS-PAGE, transferred to nitrocellulose membranes, and blotted using antibodies against Prosα6. Membranes were blocked in TBST (Tris 50mM, pH 7.2, NaCl 150mM, 0.1% Tween 20) containing 5% dry skimmed milk for 1h. Primary antibodies were diluted 1:1000 (Anti-Prosα6, described above) or 1:1000 (Anti-β-Actin, Santa Cruz #81178) in the same buffer and incubated with the membranes for 14-16h. The membranes were washed 3x for 5 min and then incubated with the secondary antibodies (Goat Anti-Rabbit IgG H&L HRP, AbCam #ab6721) diluted 1:20000 for 1h. After washing, the membranes were revealed using the Pierce ECL Western Blotting Substrate. The intensity of the bands was analyzed by densitometry with ImageJ software version 1.50i.

### *In vitro* proteasome activity assay

The proteasomal activity was measured using the Proteasome 20S Activity Assay Kit (Sigma-Aldrich (catalog #MAK172). The *in vitro* proteasome kit uses the fluorogenic peptide Leu-Leu-Val-Tyr-R110 (LLVY-R110) to measure the chymotrypsin-like proteasome activity, which has been reported to be the most specific substrate to measure proteasome activity [78]. Fat bodies and ovaries were dissected and homogenized using a glass/Teflon potter Elvehjem homogenizer in 200 μl of the following buffer (10 mM Tris/HCl, 1.1 mM $MgCl_2$, 10 mM NaCl, 0.1 mM, 1 mM, 10% (v/v) glycerol, pH 7.0). The homogenates were centrifuged at 4°C at 20,000 *g* for 10 min. 30 μg of total protein from the supernatants were used for the assays. Activities were recorded for 60 min at 37°C. The fluorescence intensities ($\lambda_{ex}$ 480 nm, $\lambda_{em}$ 520 nm) were measured at 10 min intervals under a multifunctional microplate Spectra Max M5.

### Determination of triacylglycerol (TAG) content

The abdominal fat body and ovary were dissected from control and silenced females. Each replicate (N = 4) was performed using the whole abdominal fat body from one individual insect per treatment. The organs were washed in cold 0.15 M NaCl and individually homogenized in a potter Elvehjem homogenizer in 200 μl of cold PBS. According to the manufacturer's instructions, homogenates were then subjected to enzymatic TAG determination using Triglicérides 120 kit (Doles Reagents).

### Nile Red staining of the lipid droplets (LDs)

Fat bodies were obtained from dsRNA-treated females (at least three females from each condition), on the seventh day after feeding and stained with Nile Red and DAPI, as previously described for *R. prolixus* LD analysis [79]. The fat bodies were incubated for 15 min in 1 mg/ml Nile Red and 2 mg/ml DAPI made up in 75% glycerol. Tissues were mounted in 100% glycerol and immediately imaged in a Zeiss Elyra PS.1 super-resolution structured illumination microscope. The excitation wavelengths used were 543 nm for Nile Red and 280 nm for DAPI. The peripheral regions of the fat bodies were analyzed. The average diameters of the lipid droplets were obtained from two images from each group, in three independent experiments, using the DAIME image analysis software after edge detection automatic segmentation [80]. The lipid droplets' diameters were plotted in a frequency histogram (bin width = 2).

### Hemolymph extraction and SDS-PAGE

The hemolymph was extracted from control and silenced females, 7 days after the blood meal, as originally described by Masuda e Oliveira, 1985 [81]. Approximately 10 μl of hemolymph per female was obtained by cutting one of the insects' legs and applying gentle pressure to the abdomen. The hemolymph was collected using a 10 μl pipette plastic tip. Once collected, the hemolymph was diluted 2x in PBS containing approximately 8 mg of phenylthiourea. The equivalent of 1 μl of hemolymph was loaded in each lane of a 10% SDS-PAGE.

### Scanning electron microscopy (SEM)

Freshly laid (0-24h) eggs were carefully collected and fixed by immersion in 2.5% glutaraldehyde (Grade I) and 4% freshly prepared formaldehyde in 0.5 M sodium cacodylate buffer, pH 7.3. As previously described, samples were washed in cacodylate buffer, dehydrated in an ethanol series, and coated with a thin layer of gold [82]. Models were observed in a Zeiss EVO 10 scanning electron microscope operating at 10kV.

### Statistics

Results were analyzed by Student's t-Test or chi squared test for the comparison of two different conditions and One-way ANOVA followed by Tukey's test for the comparison among more than two conditions. Differences were considered significant at $p < 0.05$. All statistical analyses were performed using Prism 7.0 software (GraphPad Software).

## Results

### The mRNAs of the 20S proteasome α- and β- subunits are maternally accumulated in the mature oocytes of *R. prolixus*

Seventeen genes encoding the proteasome conserved domain (PF00227) were identified in the *R. prolixus* genome (version C3.3 Vector Base). A phylogenetic analysis across different insect species revealed that *R. prolixus* predicted 20S proteasome genes are mostly distributed in single isoforms of each of the seven different α- and β-subunits of the 20S core proteasome (Fig 1, S2 Table), except α-subunit 2, which presented three isoforms in the genome (RPRC006558/RPRC009301/RPRC003608, greenish blue branch) and one extra uncharacterized sequence (RPR009302, black branch), which did not group with any of the known α- and β-subunits of the analyzed insect's species.

Within the *R. prolixus* chorionated (mature) oocyte transcriptome [50], only five of the seventeen genes identified above in the genome were not detected: The three isoforms of α-subunit 2, β-subunit 5, and the uncharacterized sequence (RPR009302) (S3 Table). Although the transcriptome was performed using unmated females, this finding indicates that the majority of the 20S proteasome's functional components are maternally accumulated in the oocytes, presuming a significant role for proteasomal degradation during oogenesis and/or early development.

### Proteasomal activity can be detected in the fat body and ovary of vitellogenic females

Specific proteasomal activity was detected using the fluorogenic peptide Leu-Leu-Val-Tyr-R110 (LLVY-R110) as a substrate in the fat body and ovary of adult *R. prolixus* females, as previously reported [48], indicating the presence of functional 20S proteasomes within those organs under vitellogenic conditions. Interestingly, the ovary presented approximately 50% of the activity detected in the fat body. Spontaneous substrate degradation during the assay was also monitored (in the absence of tissue homogenates) and labeled as the control trace (Fig 2).

### The 20S proteasome α-subunit 6 (Prosα6) is highly expressed in the ovary and developing oocytes

It is known that correct proteasome assembly depends on the ordered addition of each of the α- and β-subunits to the 20S α-β-β-α barrel. Thus, an induced dysfunction in at least one of those subunits is likely to result in defective proteasome activity [22,26]. To further investigate the role of the proteasomal activity during vitellogenesis and oogenesis in *R. prolixus*, we explored the expression pattern and performed RNAi functional studies targeting the Prosα6 (RPRC007220), which, among all 20S proteasome subunits, presented the highest levels of reads per kilobase of transcript per million reads (RPKM) detected in the mature oocyte transcriptome (S3 Table) [50].

The one copy of the gene α-subunit 6 (hereafter named Prosα6), in the *R. prolixus* genome assembly (RproC3), presents a total of 5 exons and encodes a predicted protein 30% similar to its human ortholog (Gene ID: 5687) including the expected 184-187-amino acid proteasome conserved domain (Pfam: PF00227) (Fig 3A). RT-qPCR showed that the ovary of *R. prolixus*

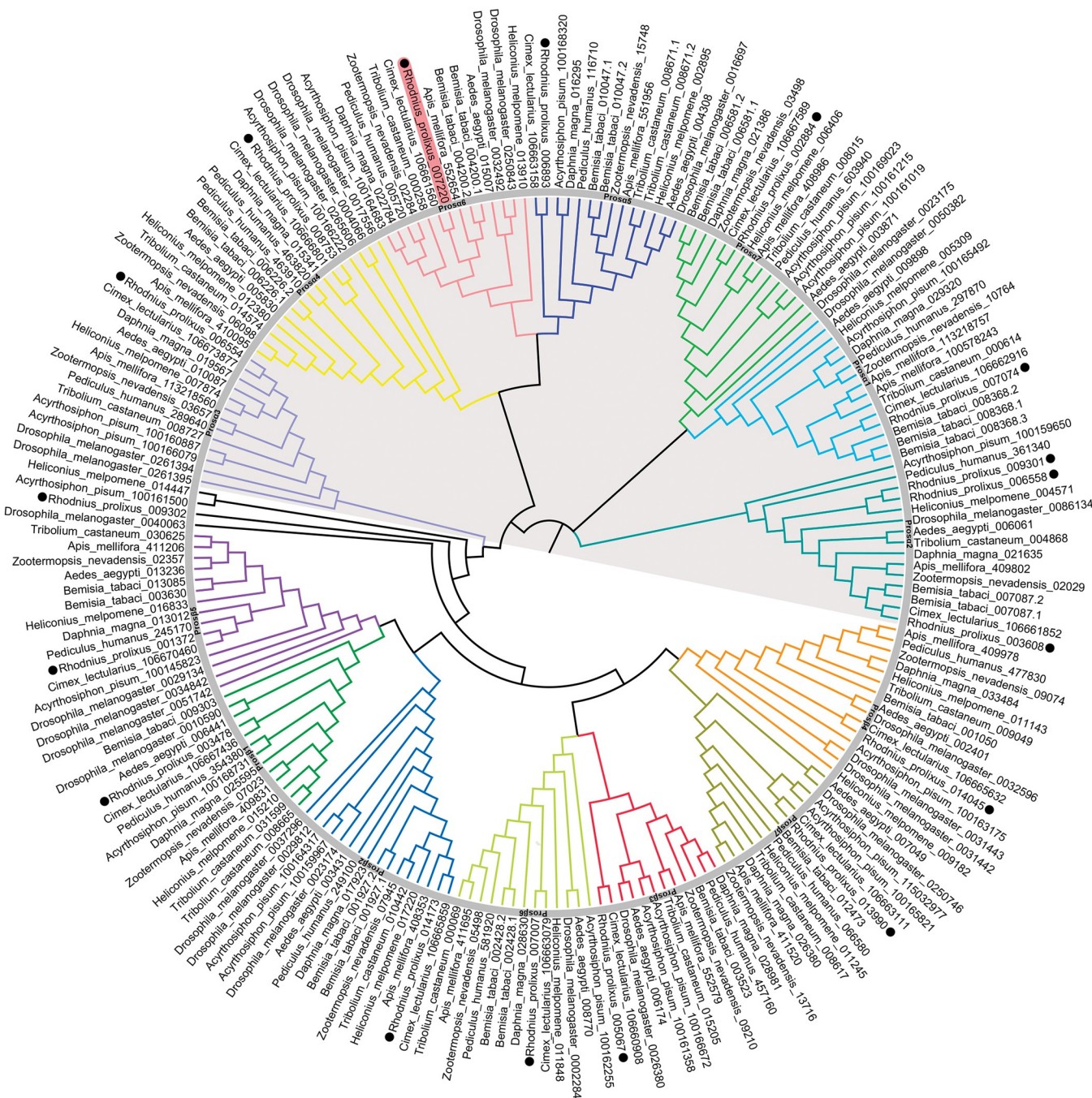

**Fig 1. Maximum likelihood phylogenetic analysis of proteins with PF00227 domain across species.** Sequences were aligned using MUSCLE tool and the phylogenetic tree reconstruction was made using maximum likelihood method under the best model characterized *Drosophila melanogaster*. The tree was designed with 1000 replicates of rapid bootstrap statistics. Each color represents one proteasome subunit α or β. Shaded gray indicates subunits α. All gene IDs are listed in S2 Table. All 20S proteasome *R. prolixus* subunits are dot-marked. *R. prolixus* Prosα6 is red-shaded.

vitellogenic females expresses an average of 4x more Prosα6 than the midgut and fat body, the other major organs of the adult insect (Fig 3B). Within the ovary, the Prosα6 mRNA is detected in the tropharium (a structure where the germ cell cluster and the nurse cells are located) and in all stages of the developing oocytes (pre-vitellogenic, vitellogenic and chorionated) (Fig 3C).

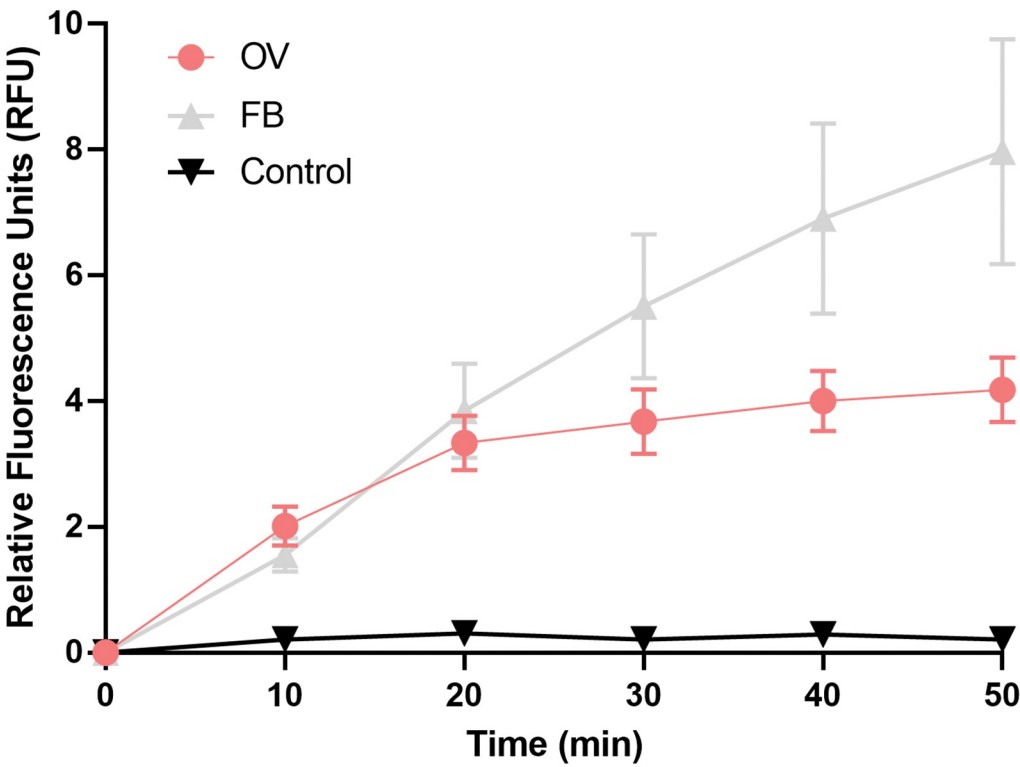

**Fig 2. Specific proteasomal activity in the ovary and fat body of vitellogenic females.** *In vitro* specific proteasomal activity, using the fluorogenic peptide Leu-Leu-Val-Tyr-R110 (LLVY-R110) to measure chymotripsin-like proteasome activity, was detected in freshly dissected ovaries and fat bodies of vitellogenic females. The activity was quantified over 50 min assays at 37°C. Spontaneous fluorogenic substrate degradation was monitored over the assay in blank reactions and shown as the control (black) trace. Each experiment was performed using the organs dissected from one individual (n = 5).

## Pros6 is effectively knocked down by RNAi in vitellogenic females, which results in less proteasomal activity in the fat body and ovary

To learn more about how vitellogenesis and oogenesis are impacted by the disturbance of proteasomal activity, we synthesized a specific double-stranded RNA designed to target the sequence of *R. prolixus* Prosα6 and delivered it to adult females by directly injecting into the insect's hemocoel two days before the blood meal. RT-qPCR showed that the knockdown of Prosα6 was systemic and efficient seven days after the blood feeding, with an average of 90% mRNA silencing in the midgut and fat body and approximately 40% mRNA silencing in the ovary (Fig 3D). Immuno-blottings demonstrated that the levels of Prosα6 protein were approximately 80% decreased in silenced ovaries when compared to controls (dsMal) (Fig 3E). To test if the silencing of Prosα6 resulted in deficient proteasomal activity, fat body and ovary samples of control and silenced females were tested for *in vitro* proteasomal activity. We found that the silencing of Prosα6 resulted in approximately 50% reduced levels of proteasomal activity in both organs (Fig 3F).

## Prosα6 knockdown does not alter lifespan, blood digestion, TAG accumulation in the fat body or general yolk protein availability in the hemolymph

The silencing of Prosα6 did not result in major alterations in the insect's general lifespan, (median survival of 28,5 days for control females and 24 days for silenced females, p>0.05

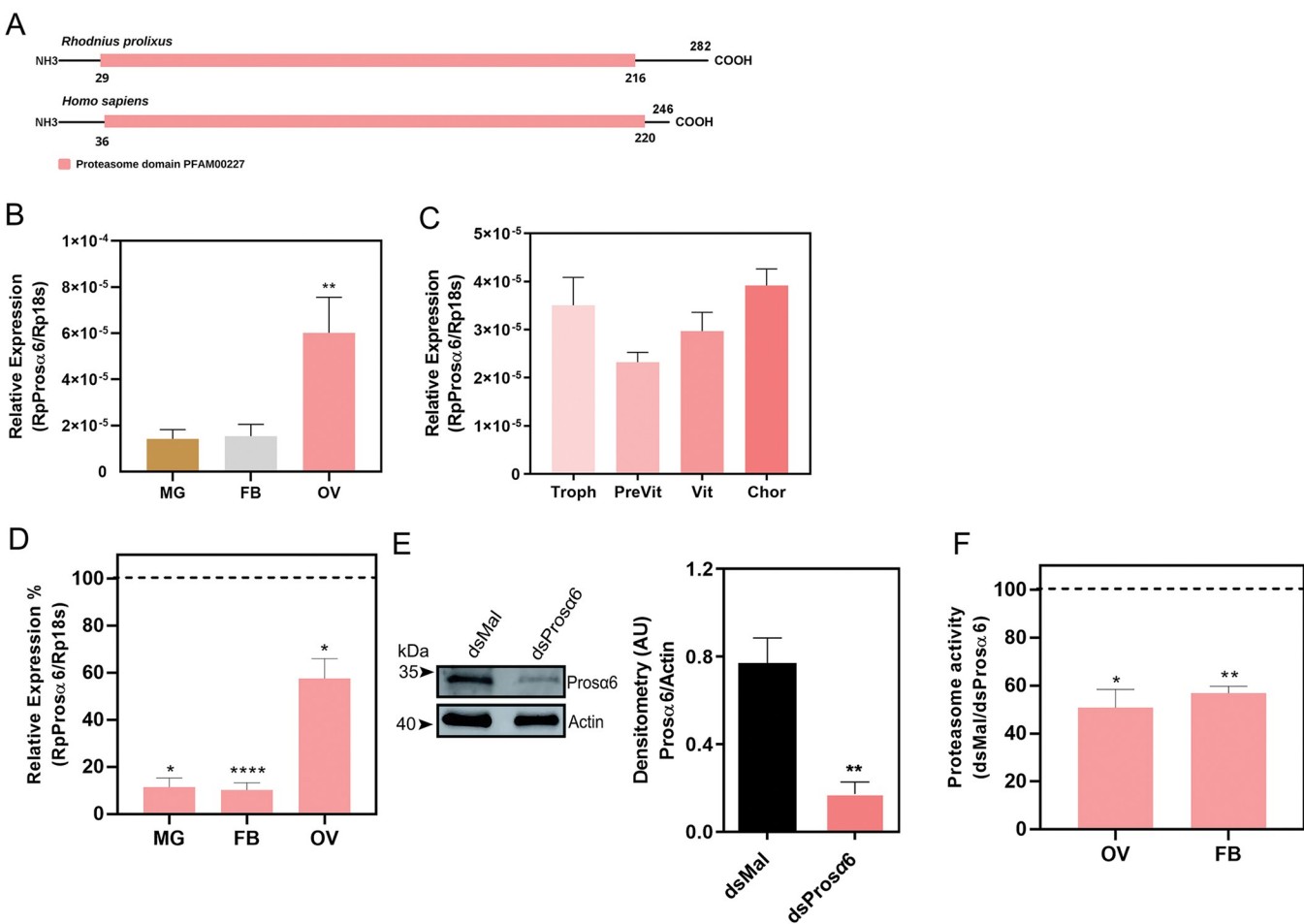

**Fig 3. Prosα6 is highly expressed in the ovary of vitellogenic females and its RNAi knockdown results in reduced proteasomal activity. (A)** Schematic diagram of the predicted conserved functional domain (proteasome domain Pfam00227) present in Prosα6 proteins from *R. prolixus* (RPC007220) and *Homo sapiens* (NCBI ID: 6587); PSMA6 NCBI ID: 6587. **(B)** RT-qPCR showing the relative expression of *R. prolixus* Prosα6 in different organs of vitellogenic females dissected 7 days after the blood meal. MG, Midgut; FB, Fat body; Ov, Ovary. **(C)** RT-qPCR showing the relative expression of Prosα6 within the ovary. Troph, tropharium; PreVit, previtellogenic oocytes; Vit, vitellogenic oocytes; Chor, chorionated oocytes. The relative expression was quantified using the ΔCT method with 18S as endogenous control. Graphs show mean ± SEM (n = 5). **p<0.01, One Way ANOVA. **(D)** RT-qPCR showing the relative expression of Prosα6 in different organs in control and silenced samples. The control (dsMal) is represented by the dotted line, and each organ percentage expression is represented by the bars. Graphs show mean ± SEM. Each experiment was performed using samples from a pool of 2 or 3 insects (n = 5) *p<0.05, ****p<0.0001, t-Test. **(E)** Prosα6 immunoblotting and densitometric quantification in control and silenced samples. Each experiment was performed using samples from a pool of 2 or 3 insects (n = 3). **p<0.01, t-Test. **(F)** *In vitro* proteasomal activity tested in the ovaries and fat bodies of control and Prosα6-silenced samples. The control (dsMal) is represented by the dotted line and each organ's activity is represented by the bars. Each experiment was performed using the organs dissected from one individual (n = 5) *p<0.05, **p<0.01, t-Test.

Log-rank (Mantel-Cox) test) (Fig 4A). Blood meal digestion, measured by quantifications of the insect's midgut protein on different days after blood feeding, presented a tendency of inhibition, and was significantly affected at 14 days after feeding (Fig 4B). Although digestion was impaired, Fig 4C shows representative images of dissected insects evidencing morphologically similar midguts (MG, red-traced lines) within the abdomens of control and Prosα6-silenced females.

To further examine the effects of Prosα6-silencing, we quantified 1) fat body triacylglycerol (TAG) contents and lipid droplets (LDs) profile, and 2) the availability of yolk proteins in the hemolymph of control and Prosα6-silenced females.

Although no changes in TAG amounts accumulated in the fat bodies of control and silenced insects were observed (Fig 4D), knockdown females presented a shift to the presence

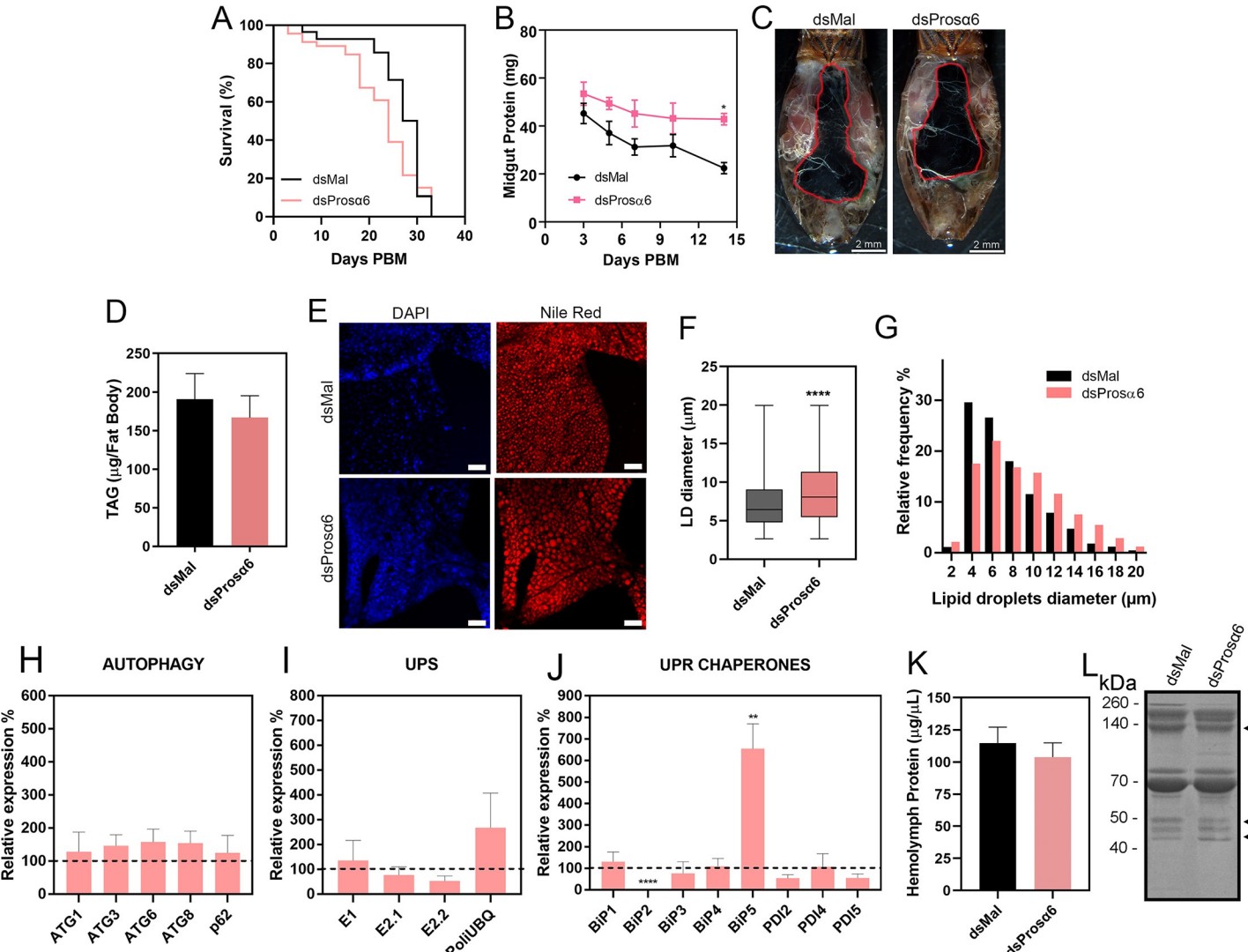

**Fig 4. Silencing of Prosα6 does not alter digestion, TAG content in the fat body and yolk protein availability in the hemolymph. (A)** Survival rates of control and silenced females (n = 45), p>0.05 Log-Rank test. **(B)** The total amount of protein was quantified in the midguts of control and Prosα6-silenced insects at different days after the blood meal to access their digestion rates. Each experiment was performed using the organs dissected from one individual (n = 8). **(C)** Representative images of the dorsal view of dissected insects pointing to the similar morphology of the midgut (MG, red-traced line). **(D)** Total TAG contents were determined in the fat bodies of control and Prosα6-silenced females. Each experiment was performed using the organs dissected from one individual (n = 5). **(E)** Lipid droplets (LDs) from freshly dissected fat bodies were stained with Nile Red, observed under the laser scanning confocal microscope. DAPI-stained nuclei are also observed. Each experiment was performed using the organs dissected from one individual (n = 3). Bars: 40 μm. **(F)** Quantification of the maximum diameter of the LDs. Each experiment was performed using the organs dissected from one individual (n = 2) and 2 images from each experiment were quantified. Graph shows median ± SD of at least 1500 LD per condition. ***p<0.001, t-Test. **(G)** Diameter distribution histograms of the LDs observed in **(E)** and **(F)**. **(H)** Expression levels of autophagy related genes in the fat body of Prosα6-silenced females. **(I)** Expression levels of UPS related genes in the fat body of Prosα6-silenced females. **(J)** Expression levels of the ER chaperones BiPs and PDIs in the fat body of Prosα6-silenced females. The control (dsMal) is represented by the dotted line, and each gene percentage expression is represented by the bars. Graphs show mean ± SEM. Each experiment was performed using samples from a pool of 2 or 3 insects (n = 4) **p<0.01, ****p<0.0001, t-Test. **(K)** Total protein in hemolymph from control and Prosα6-silenced females measured by the Lowry method. Graph shows mean ± SEM. Each experiment was performed using the organs dissected from one individual (n = 4). **(L)** 10% SDS-PAGE showing the protein profile of the hemolymphs extracted from control and Prosα6-silenced females. Arrowheads point to the vitellogenin subunits (n = 4).

of larger LDs in the fat body, as detected by Nile Red staining and quantifications (Fig 4E–4G). Despite the large internal variation in the LD diameters (Fig 4F), the fat bodies of silenced females presented higher proportions of LDs ranging from 10 to 20 μm, whereas fat bodies of control females had a higher prevalence of smaller LDs, with a diameter of 4–6 μm (Fig 4G).

These findings suggest that the dynamics of TAG accumulation and/or mobilization in the LDs were somehow impacted by the reduced proteasomal activity.

It has been extensively described that UPS and autophagy coordinate general intracellular protein degradation and general cellular proteostasis [16,83], thus we asked whether the inhibition of proteasomal activity in Prosα6-silenced fat bodies would trigger changes in the expression of the ubiquitin-conjugating enzymes and autophagy-related genes (ATGs). We found that the silencing of Prosα6 did not trigger major expression modulations in members of the autophagy pathway (ATG1, ATG3, ATG6, ATG8, and p62/SQSTM1) (Fig 4H) or the ubiquitin-conjugating system (E1, E2.1, E2.2, and polyubiquitin) (Fig 4I) in the fat body.

We also examined the expression of the ER chaperones HSP70/BiPs and PDIs, which are recognized effectors induced by the activation of the unfolded protein response (UPR). Interestingly, BiP5/GRP78 displayed a 7-fold induction, consistent with an activation of the UPR. On the other hand, BiP2 expression was strongly inhibited, while all the other isoforms of BiPs and PDIs did not exhibit any changes in expression in Pros6-silenced fat bodies when compared to control samples (Fig 4J).

Furthermore, the overall hemolymph protein levels (Fig 4K) and profile (Fig 4L, SDS-PAGE, arrowheads point to the major circulating yolk protein vitellogenin (Vg) [81]), were similar in control and Prosα6-silenced females, demonstrating that the availability of yolk proteins to be collected into the oocytes during vitellogenesis was not affected in the silenced insects.

## Prosα6 knockdown culminates in full oogenesis arrest

Even though the availability of yolk proteins in the hemolymph was not compromised, the oviposition rates of Prosα6-silenced females were severely reduced (Fig 5A). Most of the silenced females did not lay any eggs (Fig 5B), and the majority of the few laid eggs did not develop into embryos. In total, we observed an 80% reduction in the F1 hatching rates (Fig 5C). According to the compromised oviposition rates, Prosα6-silenced ovaries were fully underdeveloped, presenting only a few morphology-defective oocytes, contrasting to the standard reddish oocytes present in the ovaries of control insects (Fig 5D), indicating a dysfunction at the early stages of oogenesis. The ultrastructure of the few eggs laid by silenced insects was also observed using scanning electron microscopy, and we found that most of them presented massive alterations in the outer surface of their eggshell (Fig 5E, right panel). In some of them, the biogenesis of the operculum, a specialized anterior domain of the eggshell in *R. prolixus* (arrowheads), was completely defective (Fig 5E, left panel).

Because the uptake of the main yolk protein Vg by the oocytes is carried out by the vitellogenin receptor (VgR), we asked whether or not its expression was altered in Prosα6-silenced ovaries. We found that in control insects VgR expression presents a tendency of elevation in the ovaries on days 5 and 7 after feeding. Interestingly, in silenced ovaries, VgR expression presented a similar profile but displaying higher expression levels than control samples on days 5, 7 and 10 after feeding (Fig 5F).

Regarding the modulation of UPS, autophagy, and the UPR machinery, we found that the silencing of Prosα6 in the ovaries triggered a 300-fold induction in the expression of the adaptor protein p62/SQSTM1, whereas ATG1, ATG3, ATG6 and ATG8 presented comparatively moderate tendencies of up-regulation (Fig 6A). Regarding the ubiquitin enzymes, the silencing of Prosα6 resulted in approximately 2- to 3-fold induction of the E1-activating enzyme and the two isoforms of E2-conjugating enzymes (E2.1 and E2.2), whereas the expression levels of polyubiquitin were not altered (Fig 6B). Interestingly, except for BiP5/GRP78, which expression was not changed, all isoforms of BiPs and PDIs displayed a tendency of downregulation

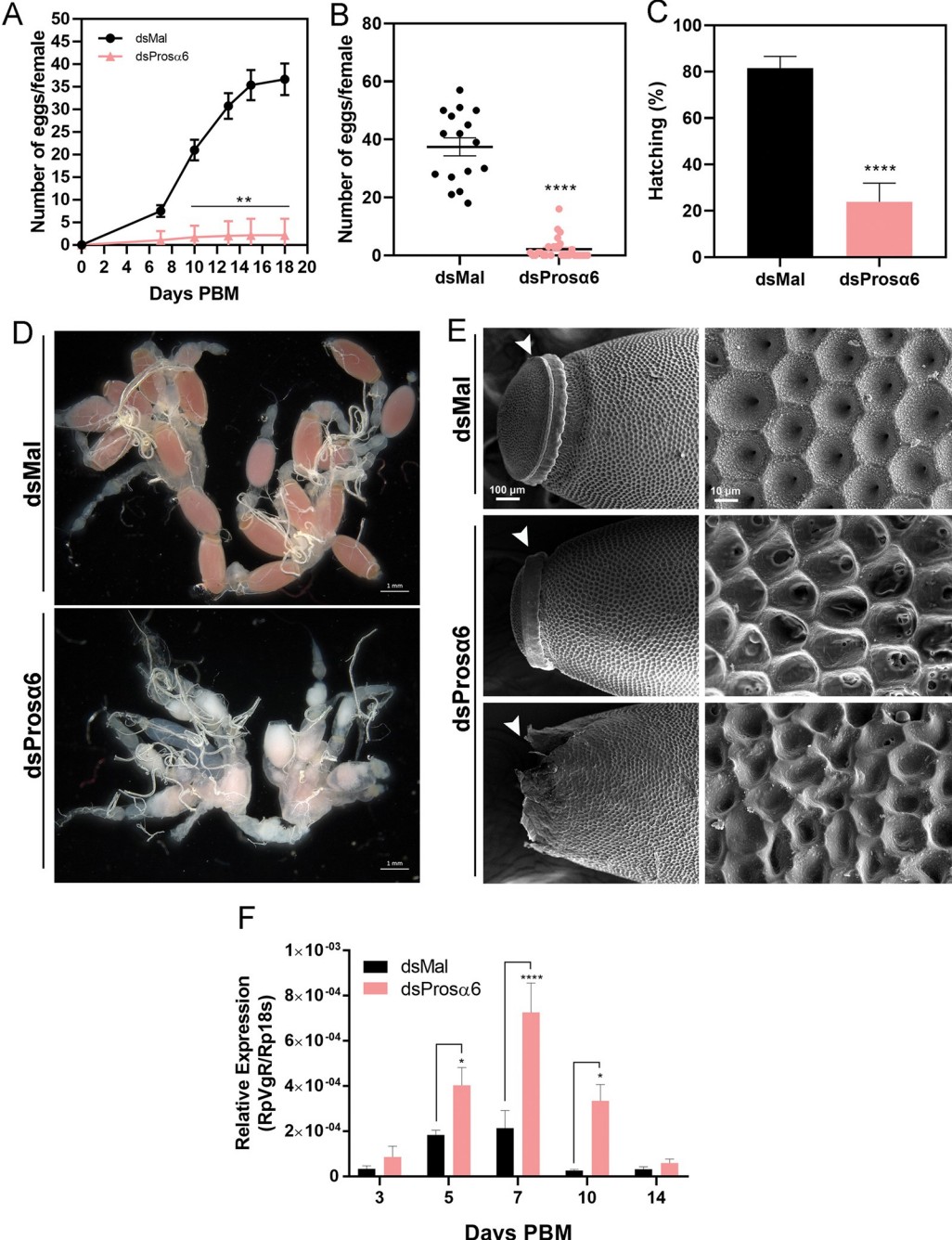

**Fig 5. Oogenesis is arrested in Prosα6-silenced insects. (A)** Number of eggs laid per female over 3 weeks after the blood meal (n = 16–20). **p<0.01 Two-Way ANOVA. **(B)** Total number of eggs laid by Prosα6-silenced and control females (n = 16) ****p<0.0001, t-Test. **(C)** Percentage of hatching of the F1 embryos in control and Prosα6-silenced insects (n = 16–20) ***p<0.0001, chi squared test. All graphs show mean ± SEM. **(D)** Representative image of dissected reproductive systems (ovaries, calyx and oviducts) from control and Prosα6-silenced females (n = 4). Insects were dissected 7 days after the blood meal. **(E)** Scanning electron micrographs of the phenotypes observed in the few F1 eggs laid by control and silenced females. Approximately half of the few eggs laid by Prosα6-silenced insects presented massive aberrations in the operculum (arrowheads, left panel) and in the outer surface of the eggshell (right panel). Images are representative of 6 eggs laid by different insects (n = 6). **(F)** RT-qPCR showing the relative expression of VgR in the ovary of control (dsMal) and silenced insects. The relative expression was quantified using the ΔCT method with 18S as endogenous control. Graphs show mean ± SEM. Each experiment was performed using samples from a pool of 2 or 3 insects (n = 4–6). * p<0.05, ***p<0.001, Two Way ANOVA.

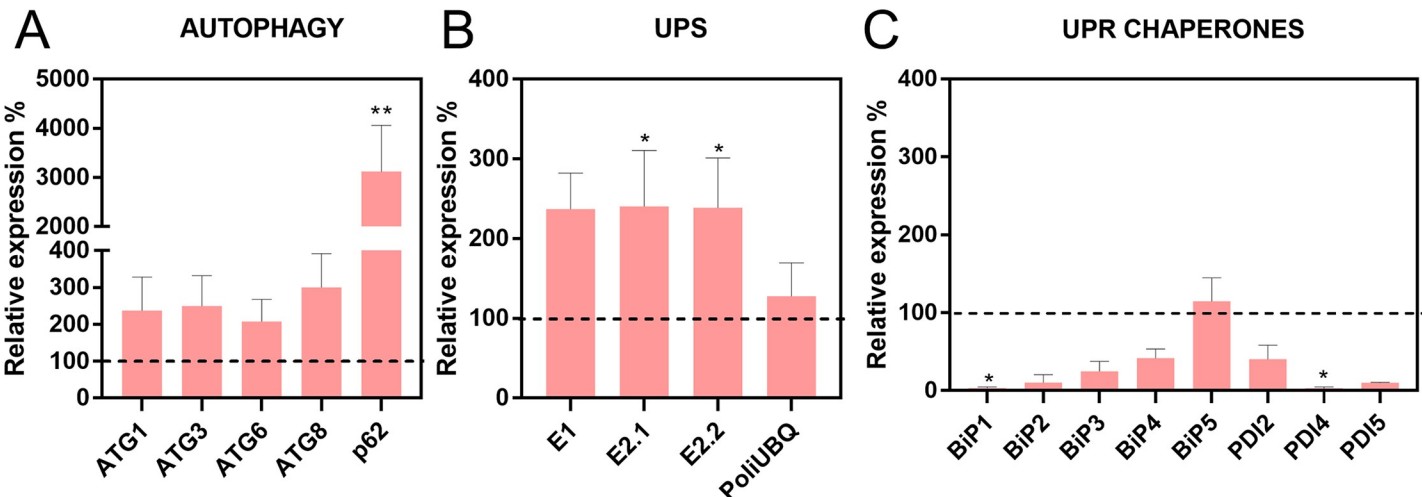

**Fig 6. Levels of autophagy, UPS and UPR related genes in Prosα6-silenced insects.** **(A)** Expression levels of autophagy related genes in the ovaries of females silenced for Prosα6. **(B)** Expression levels of UPS related genes in the ovaries of females silenced for Prosα6. **(C)** Expression levels of the ER chaperones BiPs and PDIs in the ovaries of females silenced for Prosα6. The control (dsMal) is represented by the dotted line, and each gene percentage expression is represented by the bars. Graphs show mean ± SEM. Each experiment was performed using samples from a pool of 2 or 3 insects (n = 4) *p<0.05, **p<0.01, t-Test.

in Prosα6-silenced ovaries (Fig 6C), demonstrating that although the UPS and autophagy machinery were modulated, these changes did not trigger the upregulation of the ER chaperones (a signal of disturbed proteostasis and ER stress) in the ovaries.

Altogether, we discovered that proteasomal activity is crucial for the ovary to activate egg maturation. Additional research focusing on the location and control of proteasomal activity within the ovaries may shed information on the molecular processes necessary for the progression of the oogenesis program in this model.

## Discussion

Insects' high reproductive outputs play a major role in their capacity to fill nearly every niche in nature and serve as carriers of various vector-borne diseases. To create novel strategies for vector population control, it is strategically important to identify molecular targets with biotechnological potential through the functional research of genes that are crucial for insect reproduction [84]. This is particularly important for the attempts to manage neglected diseases spread by vectors that are common in developing nations, such as dengue fever and Chagas disease (www.who.int/neglected diseases/en/).

While proteasome activity is regarded as widely conserved and ubiquitous, functional investigations have shown that proteasomes have a specific role in the reproduction biology of several models [35,42,47,49], particularly in the signals that trigger oocyte maturation [36–40,43]. In this study, we discovered that the vitellogenic ovaries exhibit significant proteasomal activity, and that the silencing of the essential 20S proteasome subunit Prosα6 causes reduced proteasomal activity, which ultimately leads to complete oogenesis arrest and reproduction ablation in the Chagas disease vector *R. prolixus*.

It is interesting to point that during oogenesis the developing oocyte is undergoing meiosis, and its pronucleus is mostly transcriptionally inactive. Most of its accumulated mRNA is maternally derived, synthesized by the nurse cells. In a meroistic-telotrophic type of ovary, such as in *R. prolixus*, the accessory nurse cells are located in the tropharium being connected to the oocytes through cytoplasmic bridges (nutritive cords) allowing the transport of

macromolecules, mRNA, and protein [12]. Thus, it is probable that the high levels of Prosα6 mRNA found in the tropharium are targeted for delivery to the developing oocytes.

Our results show that the proteasomal activity found in the ovary is approximately 50% lower than the activity detected in the fat body. However, given that both experiments were carried out using whole-organ homogenates that had been adjusted to identical levels of total protein (30 μg), one might interpret these data cautiously. In vitellogenic ovaries, the yolk proteins vitellin (VT) and Rhodnius heme binding protein (RHBP) account for more than 80% of the total proteins in the tissue [13,85]; as a result, the availability of all other non-yolk proteins in the ovary is significantly lower than in the fat body. In light of this, it is very likely that proteasomal activity in the ovaries was much underestimated. Thus, even though it is challenging to define causality in this type of experiment, our findings that proteasomal activity within the ovary is an important mechanism for the stimulation of oogenesis cannot be disputed. Additional investigations on the molecular mechanisms that are being affected by the ovary-reduced proteasomal degradation, particularly observations on the progression of oocyte meiotic maturation, will certainly enlighten our knowledge regarding this vector reproduction biology.

Regarding the fat body, while no changes in TAG reservoirs were observed in silenced insects, the 7-fold specific induction of BiP5/GRP78 among the ER chaperones, *per se*, points to the presence of an active stress signal—probably due to its reduced proteasomal activity [86,87]. In addition, the observed changes in the LDs profile (a shift to larger LDs in silenced samples) also point to some metabolic alteration occurring in this organ. It is described in the literature that the number and size of LDs can be altered under varied levels of lipolysis and lipogenic states, serving as crucial nodes of cellular metabolism by facilitating coordination and communication between various organelles and enzymes [88]. The morphologically dynamic character of the LDs in the fat body of *R. prolixus* over the gonotrophic cycle and under different conditions of molecular perturbations has been documented in a number of investigations [89–91]. Thus, it is plausible that the shift to larger LDs in the fat body of Prosα6-silenced insects is related to metabolic changes imposed by the *in situ* reduced proteasomal activity and/or to the altered integrated endocrine signals arising from the ovary under an oogenesis arrest, the later matter will be further discussed in the next session.

The fact that we discovered the same levels of total proteins in the hemolymphs of control and silenced insects is another result that merits attention. This finding suggests that the steady state availability of yolk proteins to be incorporated by the developing oocytes is comparable in both circumstances, even though blood digestion was also partially compromised in silenced insects. Nevertheless, in silenced insects, oogenesis is not triggered and the proteins available in the hemolymph do not find an uptake flow through the ovary. Therefore, it is likely that a halted flux of yolk protein synthesis and secretion to the hemolymph is occurring in the fat body of Pros6-silenced insects. This data points to feedback mechanisms where the concentration of yolk proteins in the hemolymph should regulate its own synthesis and secretion by the fat body. Interestingly, in ATG6-silenced insects, where the endocytic capacity of developing oocytes is affected, an accumulation of yolk proteins was observed in the hemolymph of vitellogenic females. In that case, however, oogenesis was not inhibited. The oviposition rates were comparable between control and silenced insects, but the generated eggs accumulated only 10–20% of the VT and RHBP observed in control eggs [92]. Altogether, these findings point to the existence of complex mechanisms regulating vitellogenesis in the fat body and Vg uptake.

Interestingly, we found that that the VgR is more expressed in Prosα6-silenced ovaries when compared to controls. Although specific conclusions cannot be drawn from these data, the greater VgR levels in the silenced samples can be the result of a transcriptional

compensatory mechanism in response to the delayed Vg absorption and oocyte maturation. It is possible that in the absence of endocytosis-receptor recycling, VgR resides in the plasma membrane, causing it to undergo programmed destruction, which in turn activates a transcriptional upregulation [10,93–95].

It has long been known that juvenile hormone (JH) and ecdysteroids (Ec), are the main hormones that regulate female insect reproduction [10,11]. In *R. prolixus*, Vg synthesis in the fat body and ovarian absorption are controlled by JH, which is released by the corpora allata and is essential for egg production [96,97]. JH can cross the plasma membrane and interact with a nuclear receptor and its co-activator: methoprene-tolerant (Met) and Taiman (Tai) [95,98]. It is possible that Prosα6-silencing is affecting the signals triggered by JH to disrupt fat body vitellogenesis and/or ovary Vg uptake. Experiments investigating JH titers and its receptors would be enlightening to further explore this possibility. On the other hand, while it is known that the levels of ecdysteroids in the hemolymph peak in the adult female at 5 days following a blood meal [99], the biochemical pathway of Ec synthesis and its effects during vitellogenesis in *R. prolixus* are still not well characterized. In adult females, the ovaries are the main source of this hemolymph ecdysteroid [100], which presumably operates by binding the heterodimeric nuclear hormone receptor made up of the ecdysone receptor (EcR) and ultraspiracle protein (USP), as it has been described in numerous insect species [101]. In *R. prolixus*, it has been claimed that Ec from the ovaries may reduce Vg synthesis in the fat body [102], and rather controls ovulation and oviposition [5]. One possibility is that interference at the early stages of oogenesis (caused by the silencing of Prosα6) renders the ovary incompetent in generating Ec to maintain the signals that modulate vitellogenesis, ovulation and/or oviposition. Although the effect of Ec in the fat body vitellogenesis has not been entirely characterized in *R. prolixus*, it has been described in a number of insect taxa [10].

The most current research indicates that UPS and autophagy interact at the mechanical level and undergo compensatory mechanisms governed by intricate feedback laws. Both routes are crucial for controlling protein homeostasis and are responsible for the majority of the proteolytic activity in cells. Since UPS and autophagy use quite different processes to degrade proteins, they can be tailored to adapt to changes in cellular function and environmental factors [83]. While autophagy engulfs bigger complexes and organelles and delivers them to the lysosome for degradation, the UPS typically targets misfolded proteins that can fit into the 20S proteasome proteolytic core [16,19,83,103]. The discovery of adaptor proteins, including p62/SQSTM1, that can transport ubiquitinated proteins to the proteasome or the autophagosome is particularly significant to this topic, but the rules that assign each protein to its destiny of breakdown remain unknown [83,104]. On this matter, our findings that p62/SQSTM1 is highly induced in the ovaries of Prosα6-silenced females is particularly interesting as it indicates the eliciting of a cooperation between UPS and autophagy under conditions of proteasomal deficiency. Such regulations were observed before in *R. prolixus*, where the silencing of the E1-activation ubiquitin enzyme also resulted in the induction of p62/SQSTM1 in the ovaries [49], as well as in HEK293 cells, where it was demonstrated that proteasome inhibition triggers the induction of p62/SQSTM1 and ATG8/GABARAP [105] while the proteasome itself can be targeted for autophagic degradation [106]. Moreover, the fact that both autophagy and UPS genes showed general tendencies toward upregulation in silenced insects' ovaries further supports our findings that, although the reduced proteasomal activity was sufficient to induce the expression of autophagy and UPS genes, the elicited response was insufficient to initiate the UPR.

Finally, it's crucial to note that even though we performed *in silico* tests for potential off targets and found no significant results, it's still impossible to entirely rule out off target effects that could partially explain the reported phenotypes.

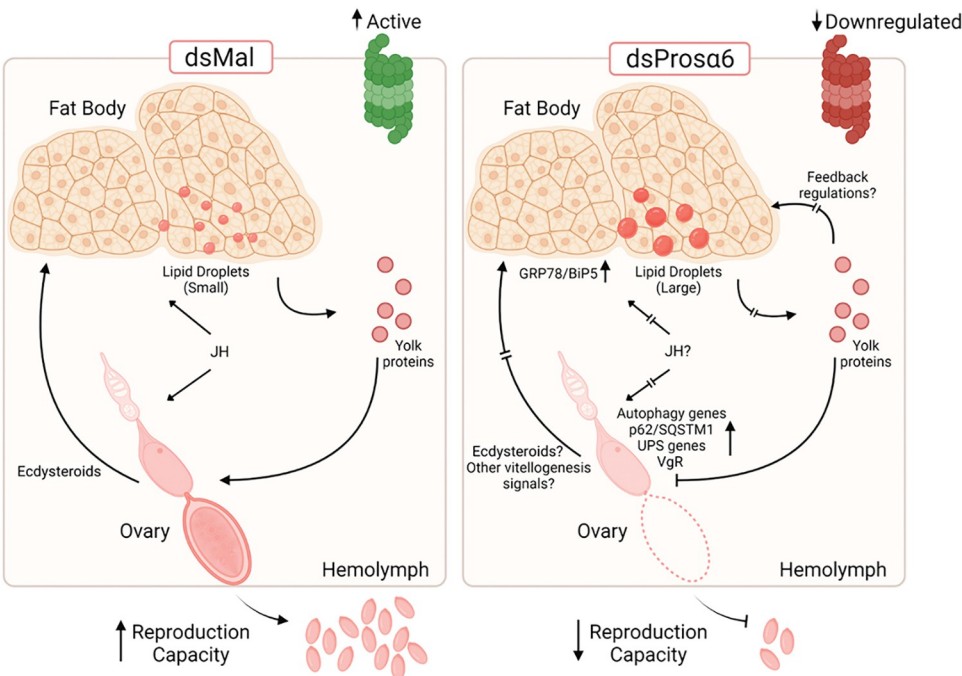

**Fig 7. Schematic diagram summarizing the effects of parental RNAi silencing of Prosα6 in vitellogenic females of *R. prolixus*.** The silencing of Prosα6 results in inhibited proteasomal activity in the fat body and ovary. Those changes result in larger LDs and upregulation in the expression of GRP78/BiP5 in the fat body, as well as a full oogenesis arrest at the early stages of oocyte maturation and upregulations in the autophagy and UPS related genes in the ovary. Because the levels of yolk proteins in the hemolymph remain unaltered under Prosα6-silencing conditions, where there is no yolk protein influx to the developing oocytes, it is possible that vitellogenesis is also arrested in the silenced fat bodies due to its *in situ* reduced proteasomal activity, feedback regulations directed by the concentration of yolk proteins accumulated in the hemolymph, the lack of signals arising from the arrested ovary to maintain an active vitellogenesis, and modulation of ecdysteroids and juvenile hormone (JH) signaling pathways.

Overall, we found that the activity of the 20S proteasome is crucial for *R. prolixus* oogenesis and highlight how additional research into the regulation of proteasome assembly and activity may shed light on the molecular mechanisms underpinning oogenesis and, subsequently, reproduction in this vector. Fig 7 displays a schematic model summarizing our discoveries and hypotheses.

## Supporting information

**S1 Fig. 18S as a stable reference gene. 18s Cts obtained from different samples and conditions.** (A) RNA extracted from the ovary, midgut and fat body. (B) RNA extracted from the different parts of the ovariole (troph, tropharium; PV, previtellogenic follicle; Vit, Vitellogenic follicle; Chor, chorionated oocyte). (C) RNA extracted from the ovary and fat body of control (dsMal) and silenced insects. All samples were dissected 7 days after the blood meal.
(TIF)

**S2 Fig. Original images of the anti-Prosα6 immunoblottings.** (A) Immunoblotting using the pre-immune serum from the immunized rabbit in control (dsMal) and silenced (dsProsα6) ovary samples. (B) Immunoblotting using the final bleed serum from the immunized rabbit in control (dsMal) and silenced (dsProsα6) ovary samples. All samples were dissected 7 days after the blood meal. A pool of 3 ovaries were used for each experiment (n = 3).
(TIF)

**S1 Table. Genes and primers List.** All sequences were obtained from *Vector Base* (https://www.vectorbase.org/) and primers were synthesized by Exxtend. T7 promoter sequence is underlined.
(DOCX)

**S2 Table. Gene IDs for all genes described in S2 Fig.**
(XLSX)

**S3 Table. 20S proteasome proteins identified in *R. prolixus* and their RPKM in the mature oocyte transcriptome.**
(PDF)

# Acknowledgments

The authors thank Danilo Oliveira for all the support, Bruna Afonso and Geane Braz for the careful care of our lab insectarium, and CENABIO-UFRJ for providing electron microscopy equipment and facilities. Fig 7 was created with BioRender.com.

# Author Contributions

**Conceptualization:** Allana Faria-Reis, Katia C. Gondim, Isabela Ramos.

**Data curation:** Allana Faria-Reis, Samara Santos-Araújo, Jéssica Pereira, Thamara Rios, David Majerowicz.

**Formal analysis:** Allana Faria-Reis, Samara Santos-Araújo, Jéssica Pereira, Thamara Rios, David Majerowicz.

**Funding acquisition:** Isabela Ramos.

**Supervision:** Isabela Ramos.

**Writing – original draft:** Isabela Ramos.

**Writing – review & editing:** Allana Faria-Reis, Samara Santos-Araújo, Jéssica Pereira, Thamara Rios, David Majerowicz, Katia C. Gondim, Isabela Ramos.

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
