## [Decision Letter · Decision Letter 0]

22 Feb 2023

Dear Dr Ramos,

Thank you very much for submitting your manuscript "Silencing of the 20S proteasomal subunit-α6 triggers full oogenesis arrest and modulation of the selective autophagy adaptor protein p62/SQSTM1 in the ovary of the vector Rhodnius prolixus" for consideration at PLOS Neglected Tropical Diseases. As with all papers reviewed by the journal, your manuscript was reviewed by members of the editorial board and by independent reviewers. In light of the reviews (below this email), we would like to invite the resubmission of a significantly-revised version that takes into account the reviewers' comments, in particular those regarding the understanding of the experiments and conclusions.

We cannot make any decision about publication until we have seen the revised manuscript and your response to the reviewers' comments. Your revised manuscript is also likely to be sent to reviewers for further evaluation.

Sincerely,

Alessandra Aparecida Guarneri

Academic Editor

Alvaro Acosta-Serrano

Section Editor

Reviewer's Responses to Questions

**Key Review Criteria Required for Acceptance?**

**Methods**

-Are the objectives of the study clearly articulated with a clear testable hypothesis stated?

-Is the study design appropriate to address the stated objectives?

-Is the population clearly described and appropriate for the hypothesis being tested?

-Is the sample size sufficient to ensure adequate power to address the hypothesis being tested?

-Were correct statistical analysis used to support conclusions?

-Are there concerns about ethical or regulatory requirements being met?

Reviewer #1: All my comments are in the "Summary and general comments section".

Reviewer #2: -Are the objectives of the study clearly articulated with a clear testable hypothesis stated? yes

-Is the study design appropriate to address the stated objectives? yes

-Is the population clearly described and appropriate for the hypothesis being tested? no

-Is the sample size sufficient to ensure adequate power to address the hypothesis being tested? yes

-Were correct statistical analysis used to support conclusions? Some parts are not clear 

-Are there concerns about ethical or regulatory requirements being met? yes

**Results**

-Does the analysis presented match the analysis plan?

-Are the results clearly and completely presented?

-Are the figures (Tables, Images) of sufficient quality for clarity?

Reviewer #1: All my comments are in the "Summary and general comments section".

Reviewer #2: -Does the analysis presented match the analysis plan? yes

-Are the results clearly and completely presented? Some of them are not

-Are the figures (Tables, Images) of sufficient quality for clarity? Some of them are not

**Conclusions**

-Are the conclusions supported by the data presented?

-Are the limitations of analysis clearly described?

-Do the authors discuss how these data can be helpful to advance our understanding of the topic under study?

-Is public health relevance addressed?

Reviewer #1: All my comments are in the "Summary and general comments section".

Reviewer #2: -Are the conclusions supported by the data presented? Not always

-Are the limitations of analysis clearly described? no

-Do the authors discuss how these data can be helpful to advance our understanding of the topic under study? Not always 

-Is public health relevance addressed? yes

**Editorial and Data Presentation Modifications?**

Reviewer #1: All my comments are in the "Summary and general comments section".

Reviewer #2: 1) Title: here, the authors mention “…modulation of the selective autophagy adaptor protein p62/SQSTM1…”, but the truth is that it was just a finding from a qPCR; in the present MS, the authors only show transcript level variations but they didn’t investigate the protein or the functional involvement of p62/SQSTM1. Thus, I consider a little ambitious to leave it in the title, for that case, a deeper study should be done.

2) Line 26: I don’t think that “genes are maternally accumulated in the transcriptome”, perhaps, transcripts accumulated into mature oocytes.

3) General speaking, the introduction is well written, there is a nice description about the UPS and the proteolytic complex 26S proteasome. I would like to read why the authors chose to study 20S and not 19S sub-complex. Also, I really believe that an introduction on Rhodnius reproductive physiology is missing since the entire MS is based on events that happen over a female reproductive cycle. 

4) Line 52: reference 2 is not correctly written in “References” section 

5) Line 68: double space 

6) Line 123: How the authors know that females were mated? Why the authors prefer females from the second or third blood feeding for experiments? Please, add to the MS your responses. Also, indicate how much the females are fed because for a full egg production it must be a minimum of times by weight.

7) Line 124: it seems that all dissections were always carried out on day 7 post blood meal and not on different days. Also, why did the authors choose 7 d PBM for experiments? 

8) Line 135: ovarian follicles

9) Line 136: …in phosphate-buffered saline (PBS, 137 mM NaCl, 2.7 137 mM KCl, 10 mM Na2HPO4, and 1.8 mM KH2PO4, pH 7.4) using…

10) Lines 141, 142 and others: according to Author_name et al. [X]

11) Line 144: Total RNA

12) Line 145 TRIzol 

13) Lines 156 and 160: It seems that the same primers were used to regular PCR and qPCR but using a different annealing temperature; was the primer efficiency tested with both temperatures?

14) Line 157: According to MIQE, the final acronym ‘RT-qPCR’ is used for reverse transcription quantitative real-time PCR.

15) Line 165: Add reference for the use of dCt method and give the complete name for 18S the first time that it is named. 

16) Line 168: the word “invariant”, is not appropriate, is not the correct expression for this case. 

17) Check when to use Table S1 or supplementary Table 1

18) The RNAi experiments were done lack an off-target control. The authors should mention this point in the MS.

19) Line 184: How the absence of cross-reactivity was tested? Please, add as supplementary Figure a complete image of western blot, showing controls and molecular weight of the protein of interest. 

20) Line 190: Was the Lowry method performed without adaptations? It is a method published in 1951. Did the authors use a kit or a microtechnique? Please, clarify in the MS

21) Line 195: The extraction buffer usually contains detergents to optimize the efficiency of extraction of soluble proteins. It is weird that in this case, the extraction buffer is just PBS and the authors didn’t mention any centrifugation step; is that correct? 

22) Line 202: “…the membranes were revealed…”

23) Line 218: Adult female reproductive system of R. prolixus is composed of 2 ovaries; are the authors considering just one ovary or both from an insect? Please, clarify it throughout the MS

24) For each experiment, did the authors use a pool of tissues or an individual tissue? Please, check it throughout the MS and add the information where necessary. As it is now, it’s not clear. Also, “N” usually refers to the population size and “n” usually refers to the sample size, how are the authors using in this case? 

25) Line 242: Was the hemolymph centrifugated to remove hemocytes? 

26) Line 246: Sentences should never begin a numeral; instead, the authors should try to reword the sentence.

27) Line 255: What is the basis for selecting t-test or chi squared test in this MS?

28) Line 272: where the transcriptome data is deposited? Also, you should discuss taking into account that the data from that transcriptome is from unmated females and you are using mated ones.

29) Line 296: abbreviation Prosα6 was already defined above.

30) Line 291-295: this phrase means the same that was written in lines 307-309.

31) Line 322: the low survival rate stolen my attention; how old these insects are?

32) Lines 323-324: can the authors add a reference for this assumption? In my opinion, this assumption should be taken with caution because when a female 7 d PBM is weighed, the values obtained are not only a representation of the MG content but also of eggs inside of those insects. Also, the truth is that according to Fig 4B, it seems to me that the weight is significatively higher in dsProsα6. Could the authors show the stat for each day?

33) Lines 345-346: the authors justify the analysis of UPR because of changes in the UPS and autophagy but the silencing of Prosα6 didn’t trigger any change. Please, rewrite this sentence. 

34) Line 350: and BIP2? Also, sometimes it is written BiP and other times BIP

35) Lines 353-355: Please, rewrite this sentence, it is not grammatically well written.

36) Line 376: It should be clarified when the authors draw conclusions or hypothesis on results that show non-statistically significant differences. 

37) Line 408: Please, add a reference for this statement.

38) Line 418: in line 350 it is written 7-fold

39) Line 411: The authors could discuss in this part about the transcript expression for Prosα6 being higher in ovaries with respect to other tissues. Also, to draw conclusion on Prosα6 maternally accumulated, please, discuss about insect oocytes pointing out that they are transcriptionally silent during vitellogenesis. 

40) Lines 437-438: in order to give more power to this hypothesis, the authors should show the protein content in the fat body of dsRNA-treated females. Also, it is not clear for me why if exist a feedback mechanism, circulating total proteins will be in the same amount. 

41) At the oocyte membrane, VgR binds Vg and once endocytosed, VgR is recycled back from oocyte cytoplasm to the membrane, where processes as ubiquitylation can take place. Taking it into account, the authors could test VgR transcript in ovaries of dsProsα6-treated insects and elucidate if the recycled back might be working or not. It could give a good response to the oogenesis arrest lead by these insects.

42) Line 447: here I found a point that deserves more attention from authors. In R. prolixus juvenile hormone (JH) controls both the production of vitellogenin in the fat body and its uptake by the ovary, and ecdysteroids (Ec) control ovulation. Recently a function controlling oocyte maturation was reported, but again, the main hormone involved in promoting vitellogenesis in R. prolixus females is JH. So, it was not clear how the authors came the conclusion that Ec could be involved. Also, the references given in the MS are from mosquitos, and the truth is that today we have a lot of bibliography on the endocrine system in Rhodnius, so, please, adapt the bibliography to the model used and reinforce the involvement of Ec because it is not well understood.

43) Line 806: (C)

44) Adapt “N” or “n” and “pool” or “individual” samples in all captions. Please, check the stat analysis throughout the MS.

45) Line 812: the relative expression shown in (D) is not the dCt as state in M&M. Please, indicate in the figure, M&M and caption that it is a percentage. 

46) What the criteria was to choose FB and OV for same test (Fig 2 and Fig 3F) and OV, FB and MG in others (Fig 3B and D)? 

47) Please, indicate molecular weight and add as supplementary figure the complete blot 

48) Line 813: indicate in M&M how the densitometric quantification was done.

49) Line 822: weighed 

50) Line 827: what does “2 experiments” mean? n=2?

51) Line 835: Total protein in hemolymph from control and Prosα6-silenced females measured by…

52) Line 841: it seems that was by 3 weeks. 

53) Line 842: on the graph (Fig 5B) there are just 16 points, does it mean n=16? Because in the captions is written N=45. Also, change to ****

54) The stat in Fig 5A is not clear. Also, add more numbers to the y and x axes to make it easier to interpret.

55) Line 843: It stolen my attention that there are N=45 for both groups since the eggs laid/ Prosα6-silenced females is very low; is it correct?

56) Line 844: It is not only ovaries but also the calyx and oviducts; perhaps, it is more correct to write “reproductive system. 

57) Schematic diagram: in general, it is a nice way to draw conclusions, just modify according to the revision. Also, Prosα6 is not inhibited but its transcript is downregulated.

58) Check for µL or µl.

**Summary and General Comments**

Reviewer #1: General comments

 In the work of Faria-Reis and co-workers entitled “Silencing of the 20S proteasomal subunit-1 α6 triggers full oogenesis arrest and modulation of the selective autophagy adaptor protein p62/SQSTM1 in the ovary of the vector Rhodnius prolixus”, the authors have studied the role of a proteasomal subunit in the oogenesis of the Chagas disease vector and model insect R. prolixus. To do so, the authors employed a series of bioinformatics, biochemistry as well as cell and molecular biology approaches. The work deals with a relevant issue, since the basic knowledge obtained could be of use to identify molecular targets for insect control. The manuscript is well organized and written, and their objectives as well as the experimental design are clear and straightforward. The findings are compelling and the conclusions match the reach of findings. Some minor suggestions are included above.

Specific comments

- Page 6, lines 110-111: "Daphnia magna" is misspelled. Please correct.

- Page 9, line 295: In order to facilitate the comprehension for a wider audience of readers, please state what “RPKM” stands for.

- Page 10, lines 305-317: Why some assessments are conducted in the midgut but some others are not? Please justify.

- Page 10, lines 321-327: Why did the authors not weight the midguts? That would have prevented to rely on indirect measurements such as the weight of the whole insect (Fig. 4B) or subjective ones such as morphology (Fig. 4C). 

- Page 10, line 353: The phrase “To investigate the fat body vitellogenesis capacity,” when related to the proteins observed in the hemolymph is not entirely accurate, since the result of the proteins circulating in the hemolymph is not only dependent on the fat body. It also results from the dynamic process of the protein uptake by the oocytes and other organs as well as from the different mechanisms of protein degradation. In fact, this point is well covered in the discussion. Please rewrite.

- Since there seems to be no differences in the offer of hemolymph proteins between silenced and control individuals, have the authors studied the uptake of yolk proteins in ovarian follicle? Did they measure the expression of vitellogenin receptor in the ovary? Even if they have preliminary data, this could be relevant to generate new hypothesis and/or to enrich the discussion.

Reviewer #2: Faria-Reis et al. identified genes that encode α and β subunits of the 20S proteasome in the genome of R. prolixus and, taking into account another MS, they concluded that those genes are maternally accumulated in mature oocytes. Then, they downregulated the transcript expression of one of the 20S proteasome subunits (Prosα6) as a tool to suppress 20S proteasomal activity. The results demonstrated that the downregulation of Prosα6 triggers full oogenesis arrest as well as changes in the transcript expression of adaptor protein p62/SQSTM1. Overall, Faria-Reis et al. show that proteasome activity is especially important for signals to initiate oogenesis. I consider this work contains very interesting results; however, it is important to address several points/observations to improve the understanding of the experiments and conclusions.

PLOS authors have the option to publish the peer review history of their article (what does this mean?). If published, this will include your full peer review and any attached files.

Reviewer #1: No

Reviewer #2: No
---

## [Editor Report · Decision Letter 1]

15 May 2023

Dear Dr Ramos,

We are pleased to inform you that your manuscript 'Silencing of the 20S proteasomal subunit-α6 triggers full oogenesis arrest and increased mRNA levels of the selective autophagy adaptor protein p62/SQSTM1 in the ovary of the vector Rhodnius prolixus' has been provisionally accepted for publication in PLOS Neglected Tropical Diseases.

Best regards,

Alessandra Aparecida Guarneri

Academic Editor

Álvaro Acosta-Serrano

Section Editor

---

## [Editor Report · Acceptance letter]

30 May 2023

Dear Dr Ramos,

We are delighted to inform you that your manuscript, " Silencing of the 20S proteasomal subunit-α6 triggers full oogenesis arrest and increased mRNA levels of the selective autophagy adaptor protein p62/SQSTM1 in the ovary of the vector *R. prolixus* ," has been formally accepted for publication in PLOS Neglected Tropical Diseases.

Best regards,

Shaden Kamhawi

co-Editor-in-Chief

Paul Brindley

co-Editor-in-Chief
